# Multiomics Analysis of a Micronutrient-Rich Dietary Pattern and the Aging Genotype 9p21 on the Plasma Proteome of Young Adults

**DOI:** 10.3390/nu17081398

**Published:** 2025-04-21

**Authors:** Sara Mahdavi, Katie Rosychuk, David J. A. Jenkins, Andrew J. Percy, Christoph H. Borchers, Ahmed El-Sohemy

**Affiliations:** 1Department of Nutritional Sciences, Temerty Faculty of Medicine, University of Toronto, 6 Queens Park Crescent, Toronto, ON M5S 3H2, Canada; sara.mahdavi@utoronto.ca (S.M.); k.rosychuk@mail.utoronto.ca (K.R.); david.jenkins@utoronto.ca (D.J.A.J.); 2Department of Nutrition, Harvard T.H. Chan School of Public Health, Harvard University, 677 Huntington Ave, Building B, Room 359, Boston, MA 02115, USA; 3Clinical Nutrition and Risk Factor Modification Centre, St. Michael’s Hospital, Toronto, ON M5B 1W8, Canada; 4Division of Endocrinology and Metabolism, Department of Medicine, St. Michael’s Hospital, Toronto, ON M5B 1W8, Canada; 5Department of Medicine, Temerty Faculty of Medicine, University of Toronto, Toronto, ON M5S 1A1, Canada; 6Li Ka Shing Knowledge Institute, St. Michael’s Hospital, Toronto, ON M5B 1W8, Canada; 7Genome BC Proteomics Centre, University of Victoria, Victoria, BC V8Z 7X8, Canada; apercy@isotope.com; 8Department of Applications Development, Cambridge Isotope Laboratories, Inc., Tewksbury, MA 01876, USA; 9Segal Cancer Proteomics Centre, Lady Davis Institute for Medical Research, Jewish General Hospital, McGill University, Montreal, QC H3T 1E2, Canada; christoph.borchers@mcgill.ca; 10Gerald Bronfman Department of Oncology, McGill University, Montreal, QC H3T 1E2, Canada; 11Department of Pathology, McGill University, Montreal, QC H3T 1E2, Canada; 12Division of Experimental Medicine, McGill University, Montreal, QC H3T 1E2, Canada

**Keywords:** micronutrients, 9p21 genetic variant, plasma proteomics, dietary patterns, gene–diet interaction, complement C4-γ

## Abstract

**Background**: Diet is one of the most significant modifiable lifestyle factors influencing human health, contributing to both morbidity and mortality. Genetic variations in the pleiotropic 9p21 risk locus further shape premature aging, disease susceptibility, and have been strongly linked to cardiovascular disease (CVD), metabolic disorders, certain cancers, and neurodegenerative conditions. However, given that this region was discovered based on Genome-Wide Association Studies, the mechanisms by which 9p21 exerts its effects remain poorly understood and its interactions with diet and biomarkers are insufficiently explored. **Methods**: This study investigated the association between the rs2383206 SNP in 9p21, dietary patterns, and plasma proteomic biomarkers in a multi-ethnic cohort of 1280 young adults from the Toronto Nutrigenomics and Health Study. Participants’ dietary intake was assessed using a validated food frequency questionnaire, and dietary patterns were categorized using principal component analysis. Plasma proteomics analyses quantified 54 abundant proteins involved in the cardiometabolic and inflammatory pathways. Genotyping identified individuals who were homozygous for the 9p21 risk allele (GG), known to confer the highest susceptibility risk to premature aging and multiple chronic diseases. **Results**: A significant interaction was observed between the 9p21 genotype and adherence to a micronutrient-rich Prudent dietary pattern for eight plasma proteins (α_1_ Antichymotrypsin, Complement C4 β chain, Complement C4 γ chain, Complement C9, Fibrinogen α chain, Hemopexin, and Serum amyloid P-component). However, only Complement C4-γ showed a pattern consistent with the risks associated with the 9p21 genotype and adherence to a Prudent diet. Individuals with the high-risk GG genotype had significantly higher concentrations of Complement C4-γ, but only among those with a low adherence to a Prudent diet. **Conclusions**: These findings suggest that Prudent dietary patterns rich in micronutrients may counteract genetic-mediated proinflammatory susceptibility by modulating key proteomic biomarkers in young adults, highlighting the potential for tailored dietary interventions to mitigate disease risk. This study also introduces a novel framework for post hoc micronutrient resolution within dietary pattern analysis, offering a new lens to interpret nutrient synergies in gene–diet interaction research.

## 1. Introduction

Chronic diseases, including cardiovascular disease [1] (CVD), type 2 diabetes [2], certain cancers [2], and neurodegenerative diseases [3], are leading causes of global morbidity and mortality. These diseases often begin developing years before clinical symptoms appear, making early diagnosis and intervention challenging. Inflammation plays a pivotal role in many of these diseases, contributing to the disruption of metabolic and cellular processes, which can lead to tissue damage, disease progression, and overall accelerated aging [4]. While traditional risk factors such as insulin resistance, dyslipidemia, and hypertension can help to predict future disease onset, they often fail to capture the early molecular changes that drive disease development. Therefore, targeting novel biomarkers, particularly those related to inflammation, and understanding how these biomarkers interact with genetic risk factors, could provide key insights for early prevention strategies. The pathogenesis of CVD and other chronic diseases (CDs) is linked to both non-modifiable factors such as genetics and modifiable environmental influences such as diet [4,5,6]. Inflammation plays an important role in the early stages of CVD, contributing to endothelial dysfunction, plaque formation, and the progression of atherosclerosis, even before traditional risk factors become apparent. Markers of cardiometabolic disease (CMD), such as glucose intolerance, insulin resistance (IR), dyslipidemia, hypertension, and abdominal obesity, predict higher rates of CVD events later in life [7]. However, traditional markers such as dyslipidemia [8] can fall short in predicting CVD in the absence of clinical symptoms. Thus, targeting novel biomarkers of CVD risk, including inflammatory markers in young adults [8,9,10], may help to establish early preventative strategies to reduce the burden of disease later in life.

One of the strongest genetic markers associated with CD risk is the 9p21 region [11,12,13]. Single nucleotide polymorphisms (SNPs) in the 9p21 region, which were discovered using early genomics applications and agnostic approaches to this region based on Genome-Wide Association Studies, have been linked its common variation leading to myocardial infarction [14] and CMD risk factors, such as type 2 diabetes (T2D) [15] and IR [7,15,16]. The 9p21 SNP rs2383206 is associated with CVD and proinflammatory state risks [14,17,18]. The 9p21 risk locus is in a non-coding region, hence, it does not code for any discovered proteins; its exact functional mechanisms in the pathogenesis of CVD and other CDs remain unclear. Evidence suggests this variation is involved in the regulatory region that affects gene expression downstream and may influence the pathways involved in inflammation and endothelial dysfunction, which could contribute to early pleotropic disease onset and progression [19]. We have shown that common variants in the 9p21 region are associated with elevated fasting serum insulin [7,15], in the absence of other biomarkers of CMD, in young adults from different ethnocultural groups living in Canada. This suggests that the genetic influence of 9p21 on CVD risk might be mediated through dysregulated insulin signaling, which is a known contributor to endothelial dysfunction, some forms of cancer, neurodegenerative disease, and atherosclerosis [18,19,20]. More recently, we have demonstrated that a low-inflammatory diet that is high in fruits, vegetables, whole grains, and legumes, and low in meat, processed foods, and refined carbohydrates is associated with lower fasting insulin and lower insulin resistance in the highest 9p21 genetic risk group only. This suggests that Mediterranean-style or ‘Prudent’ dietary patterns that are rich in micronutrients and plant-based antioxidants might possibility mitigate the 9p21-related genetic risk of CMD.

Given the limitations of traditional biomarkers in predicting CVD risk before the onset of clinical symptoms [21], there is a need to identify the novel biomarkers of CVD risk, particularly in young individuals with 9p21 risk variants. Recent advances in proteomics have enabled the simultaneous measurement of many different plasma proteins involved in distinct physiological pathways [21,22,23,24]. Using a targeted proteomics approach can enhance the predictive power to reveal the novel associations between plasma proteins and disease risk [25]. In addition, concentrations of certain plasma proteins have been found to be associated with aging and an increased risk of age-related diseases, such as Alzheimer’s and CVD [26,27]. Identifying a link between these proteins and dietary factors could provide a foundation for early preventative strategies for such diseases.

We have previously reported that genetic variations in 9p21 are associated with the unique plasma proteomic profiles involved in altered lipid metabolism and inflammatory pathways that contribute to potential CD risk, which may indicate alternative CD pathophysiology in young adults from different ethnocultural groups [24]. However, given that genetic risk factors are non-modifiable, it is important to investigate how those in the highest genetic risk groups can mitigate CD risk through lifestyle factors such as diet and nutrition.

The targeted proteomics analysis was conducted using a panel of 54 abundant plasma proteins, selected based on their involvement in inflammatory, metabolic, and vascular processes, and their prior successful application in large-scale population studies [23,28]. This panel’s utility was previously demonstrated in the Toronto Nutrigenomics and Health Study (TNHS) [23], where it effectively captured biologically meaningful variations related to diet, genetics, and lifestyle factors, providing internal validation for its use in the current study.

Diet, being the most diverse and modifiable environmental factor that interacts with genetic risk factors to influence CD risk [29], may provide an opportunity for the early intervention and prevention of CD, especially in high-genetic-risk groups. Moreover, dietary patterns have been shown to significantly influence plasma proteomic profiles [23], highlighting the importance of considering modifiable environmental factors in understanding and predicting disease risk. Dietary patterns, particularly those with micronutrient-rich profiles, often characterized as “Prudent”, which are high in fruit, vegetables, nuts, legumes, beans, and whole grains [15], have been associated with reduced CD risk [30]. Therefore, studying gene–diet interactions can provide a more comprehensive understanding of the pathways leading to CD [29] and, importantly, can help to identify diets that are personalized to one’s genetic risk factors using intermediary targeted plasma proteomics panels. Understanding these interactions could identify novel proteomic biomarkers for early detection and intervention in managing CD risks. The aim of this study was to investigate whether dietary patterns interacted with the 9p21 genotype, impacting the plasma proteome, in a multi-ethnic cohort of young adults living in Canada.

## 2. Materials and Methods

### 2.1. Study Population

Participants were recruited from the Toronto Nutrigenomics and Health Study (TNHS), which is a cross-sectional study designed to investigate the interactions between diet and genetics on the biomarkers of chronic diseases and which has been described in detail elsewhere [24]. This study was approved by the University of Toronto Research Ethics Board. A total of 1649 participants, aged 20–29 years, from diverse ethnocultural backgrounds, were enrolled from the University of Toronto’s St. George campus between 2004 and 2010. Ethnocultural background was self-reported by participants through an open-ended questionnaire, and the participants were subsequently categorized into four groups: Caucasian, East Asian, South Asian, or Other. The Caucasian group included individuals identifying as European, Middle Eastern, or Hispanic, while the East Asian group comprised Chinese, Japanese, Korean, Filipino, Vietnamese, Thai, and Cambodian participants. The South Asian group included individuals from Bangladesh, India, Pakistan, and Sri Lanka. The Other group consisted of those who were Indigenous, African American, or who identified with more than one ethnic group. Anthropometric data, including height, weight, waist circumference, and blood pressure, were also collected from all participants.

### 2.2. Dietary Patterns

Dietary patterns were assessed using principal component analysis (PCA) based on data collected from the validated Toronto-modified Harvard 196-item food frequency questionnaire (FFQ) [31,32]. This approach enabled the identification of distinct dietary clusters, consistent with prior analyses conducted in this cohort [15,31]. To enhance the accuracy in portion size estimation, participants received visual aids and detailed instructions on completing the FFQ. The questionnaire captured their dietary intake over the preceding month, including both food and supplement consumption. Responses were subsequently translated into estimated daily servings for each reported item.

PCA identified three predominant dietary patterns: “Prudent”, “Eastern”, and “Western” [31]. The Prudent pattern was characterized by the high intake of fruits and vegetables, nuts, dried lentils and beans, whole grains, and water. The Eastern pattern included a greater consumption of vegetables, rice, organ meats, and seafood. In contrast, the Western pattern was marked by the frequent intake of processed foods, items high in salt and sugar (e.g., chocolate, candy bars, processed meats), enriched wheat flour products such as pizza and doughnuts, and sugar-sweetened or caffeinated beverages. Each dietary pattern was then categorized into low-, medium-, or high-adherence groups based on the composite scores reflecting participants’ consumption of the target foods over the past month. Specifically, the “low” adherence group included individuals with scores at or below the 25th percentile; the “medium” group comprised those between the 26th and 74th percentiles; and the “high” group included those scoring at or above the 75th percentile.

The Toronto-modified Harvard FFQ used in this study was previously validated in a subsample of this cohort using a repeat FFQ one year later, along with a 3-day food record [32]. This validation demonstrated acceptable relative validity for a range of micronutrients. Specifically, the deattenuated Pearson correlation coefficients were as follows: vitamin A (r = 0.46), vitamin C (r = 0.41), vitamin D (r = 0.63), vitamin E (r = 0.39), folate (r = 0.47), sodium (r = 0.20), potassium (r = 0.43), and magnesium (r = 0.61). These values reflect moderate to strong agreement with the reference measures, with the exception of sodium, which is known to be underestimated in self-reported dietary assessment tools. Gross misclassification rates across these nutrients were low (mean ~5.7%), supporting the FFQ’s application for estimating habitual micronutrient intake in multi-ethnic young adult populations.

### 2.3. Genotyping

The subjects provided a 12 h fasting blood sample for DNA isolation. DNA was extracted from the peripheral leucocytes from whole blood samples using standard procedures, as previously described [33]. DNA was analyzed to identify the rs2383206 SNP in 9p21 that has been shown consistently to be associated with CVD risk [33]. Genotyping was completed using the iPLEX Gold assay with mass-spectrometry-based detection (Sequenom MassARRAY platform; Agena Bioscience, San Diego, CA, USA) at the Clinical Genome Centre at Princess Margaret Hospital, University Health Network in Toronto, Canada, as described previously [7]. Since the presence of two risk alleles in the 9p21 region has been associated with the highest risk of CVD and CDs [34], subjects were classified into two groups based on whether they had two copies of the risk allele versus having one or two low-risk alleles. The rs2383206 low-risk allele was considered “A”, and the high-risk allele was considered “G”. Hence, low-risk genotype subjects had either the AA or AG genotype. The high-risk group was defined as those with two copies of the risk alleles, homozygous for G, and was defined as the GG genotype.

### 2.4. Proteomics Analysis

The plasma proteomics panel for this cohort has been previously described in detail [24]. Briefly, plasma was obtained from 12 h fasting blood samples and was frozen at −80 °C. Frozen plasma samples were shipped to the University of Victoria Genome British Columbia Proteomics Centre in Victoria, Canada. Endogenous plasma proteins were quantified against calibration curves made from the SIS standards (a “reverse-curve” approach) spiked into a standard plasma sample. The SIS peptide concentrations were balanced to match the natural abundance of the proteins they represented. Of the 63 proteins measured and identified, 6 were below the detection limit and 3 had inter-assay coefficient of variations (CVs) of >20% and were excluded from the statistical analyses. Of the 54 proteins quantified, 50 had CVs of <10% and 4 had CVs of 10−14%. This set of 54 proteins were retained for further analyses as part of the TNHS. The concentrations for most of the 54 proteins included in the original assay were within the range of reported values or within a factor of two from the previously reported values using validated proteomics methods [35,36].

### 2.5. Statistical Analysis

R Studio v.4.4.1 was used for all analyses. The α-error was set at 0.05, and the *p*-values presented were two-sided. Variables that were not normally distributed were log-transformed before analysis to improve normality. The *p*-values from models using the transformed protein concentrations were reported, but the untransformed means and measures of spread were reported to facilitate interpretation. The 9p21 genotypes were examined for the Hardy−Weinberg equilibrium, and a Chi-square test was used to test for differences in the prevalence of 9p21 gene variants across the different ethnocultural groups. The initial study population was 1649 individuals. Those using hormonal contraceptives were excluded due to their known effects on the plasma proteome [37], as were those missing genetic, dietary, and proteomic data. Complete data on 1280 participants were included for this study’s analysis.

Using multivariate linear regression analyses, differences in mean plasma protein concentrations were examined between 9p21 SNP rs2383206 genotypes, stratified into two groups: those who had the low-risk alleles AA and AG (n = 958) and those who had two copies of the high-risk allele GG (n = 332). These analyses included the subject characteristics of age, BMI, waist circumference, systolic and diastolic blood pressure, blood glucose, insulin, CRP, LDL, HDL, and TG. Furthermore, preliminary sensitivity analyses stratified by sex and by ethnocultural background were conducted; however, these analyses revealed no significant modification of the genotype–diet–protein associations by these factors, which justified collapsing the subgroup groups for improved power in the final analyses. Differences in the mean diet scores for Prudent, Western, and Eastern diets were analyzed, as well as the differences between sexes. Analysis of covariance (ANCOVA) was used to examine if the 9p21 genotype, dietary patterns, and gene–diet interactions were associated with different plasma proteins from the proteomics panel. A two-way analysis of variance (ANOVA) was conducted to assess the effects of dietary adherence (Prudent, Western, Eastern), genotype (rs2383206), and their interaction on the concentrations of 54 proteins. In the final models, the analyses were adjusted for age, sex, physical activity, ethnicity, and body mass index. Western and Eastern dietary patterns had no significant associations and poorly defined any food consumption patterns in this study’s sub-population, and therefore were eliminated form further analyses, with only Prudent dietary patterns indicating robust adherence and non-adherence patterns for further analyses. Next, group differences were first assessed using a one-way analysis of variance (ANOVA), with independent variables defined by genotype (two-level categorical variable with unequal group sizes) and dietary pattern adherence (low-, medium-, and high-adherence tertiles). Given the unequal sample sizes across genotype strata—with the heterozygous or high-risk group comprising approximately 25% of the sample—post hoc comparisons were conducted using Tukey’s Honestly Significant Difference (HSD) test with unequal n adjustment. This approach accounted for differences in group sizes while controlling the familywise error rate during pairwise comparison. Partial correlation was used to assess the nutrient composition of the Prudent dietary patterns in post hoc analyses of the FFQ data with r indicating the slope and direction of association and significance being set at *p*-value < 0.05.

## 3. Results

The subject characteristics are summarized in Table 1 for the two groups according to rs2383206—carriers of the low-risk allele (A) versus those who were homozygous for the risk allele (G). An additive model was initially assessed, but the effects were only observed among homozygotes for the risk variant, suggesting a recessive model. As such, the final analyses showed the dichotomized groups as being those who were carriers of one or two low-risk alleles versus those who were homozygous for the risk allele. The distribution of men and women was similar between the two groups. Both groups had comparable adherence to the Prudent and Western dietary patterns; however, individuals with the GG genotype had a lower Eastern dietary pattern score (*p* = 0.03). The proportion of the different ethnic groups differed between genotype groups.

Table 2 shows the average protein concentrations by 9p21 genotype and adherence to the Prudent dietary pattern score. Upon the analysis of 54 different protein concentrations (Appendix A), 8 proteins were found to have a significant interaction with the 9p21 genotype and the Prudent dietary pattern (Table 2). Additionally, five protein concentrations were significantly associated with the 9p21 genotype alone (α_1_ Antichymotrypsin, Complement C4 β chain, Complement C4 γ chain, Hemopexin, and Serum amyloid P-component), while two proteins were significantly associated with adherence to the Prudent dietary pattern (Complement C9 and Hemopexin) (Table 2).

A detailed summary of the eight plasma proteins exhibiting statistically significant interactions between the 9p21 genotype and adherence to the Prudent dietary pattern is presented in Table 3. Among these, Complement C4-γ displayed the most robust and interpretable interaction: GG homozygotes demonstrated elevated concentrations under low dietary adherence, with levels attenuating progressively with greater adherence to the Prudent pattern. A similar directionality was noted for the Complement C4 β chain and α1 Antichymotrypsin, where differences between genotypes were primarily evident among participants with lower quality diets. Hemopexin and Serum amyloid P-component showed statistically significant interactions as well as independent associations with both genotype and dietary adherence, suggesting multifaceted sensitivity to both genetic and nutritional factors. In contrast, Fibrinogen α chain and Fibrinopeptide A presented more modest interactions, though still achieving statistical significance, with between-group variation being less pronounced. Complement C9 also demonstrated a significant interaction, though mean concentrations across strata remained comparatively stable. Collectively, these findings highlighted a panel of proteins with relevance to inflammation, vascular remodeling, and host defense pathways, whose circulating levels appeared to be modulated by the interplay of genetic susceptibility and diet quality [38,39].

Although the eight proteins listed in Table 3 yielded a significant interaction term, only one followed a pattern consistent with the risks associated with a Prudent diet and 9p21 genotype. Complement C4-γ concentrations decreased as adherence to a Prudent dietary pattern increased in those with the risk genotype (GG), whereas there was no difference in Complement C4-γ concentrations as adherence to the Prudent dietary pattern increased in those with the low-risk AA+AG genotype (Figure 1). Those with the risk genotype and low adherence to a Prudent diet had, on average, 17% higher levels of Complement C4-γ when compared to those with the low-risk genotype and low adherence to a Prudent diet (Figure 1). These gene–diet interactions’ effects on Complement C4-γ concentrations remained significant after adjusting for covariates age, sex, physical activity, and the log of body mass index.

Table 4 presents the correlations between dietary patterns and micronutrient intake, highlighting the nutrient composition characteristics of different dietary habits. Strong positive associations were observed for several essential vitamins and minerals, indicating that micronutrient-rich dietary patterns are linked to a higher intake of nutrient-dense foods. Notably, vitamin A (r = 0.57, *p* = 2.7 × 10^−45^), vitamin C (r = 0.31, *p* = 2.3 × 10^−13^), and vitamin E (r = 0.46, *p* = 1.6 × 10^−28^) demonstrated significant correlations, suggesting that dietary patterns emphasizing these vitamins are predominantly plant-based, with a higher consumption of fruits, vegetables, and whole foods. Similarly, potassium (r = 0.55, *p* = 6.8 × 10^−43^) and magnesium (r = 0.62, *p* = 3.3 × 10^−58^) exhibited strong positive correlations, reinforcing their association with dietary patterns rich in unprocessed plant-derived foods. Folate (r = 0.31, *p* = 5.7 × 10^−13^) also showed a significant positive relationship, consistent with diets that include leafy greens, legumes, and fortified grains. In contrast, dietary cholesterol (r = −0.11, *p* = 0.01) displayed a weak negative correlation, suggesting an inverse relationship with micronutrient-dense dietary patterns, while vitamin D (r = 0.07, *p* = 0.12) did not show a significant association. These findings underscore the nutrient profile of dietary patterns and suggest that adherence to micronutrient-rich diets may contribute to improved metabolic and inflammatory regulation through the increased intake of essential vitamins and minerals.

## 4. Discussion

Variations in the 9p21 gene are some of the most consistent genetic markers of CVD risk to date, but the number of studies that have assessed whether dietary factors can modify these risks in young adults is extremely limited. Despite this, the 9p21 locus remains one of the most widely replicated genetic risk factors for cardiovascular disease (CVD), with pleiotropic effects observed across multiple biological systems. In our previous work, we demonstrated that this locus is associated with traditional cardiometabolic biomarkers, including fasting insulin [38] and insulin sensitivity, highlighting its interaction with dietary patterns in modifying disease risk in young adults. Since age is a significant risk factor for CVD, studying a young adult cohort reduces some of the confounders associated with age, providing an ideal life-stage to observe gene–diet interactions that may provide mechanistic insights into disease development later in life. Our earlier findings implicating insulin-related pathways in gene–diet interactions provided the foundation for extending this work to the proteome in order to understand the potential mechanistic pathways involved. By examining how genotype and dietary patterns shape plasma protein expression, our findings offer novel mechanistic insights into the early molecular links between genetic risk, nutritional behavior, and cardiometabolic disease. We sought to investigate the association between diet, a common SNP in the 9p21 region which has been linked to CVD risk, and high-abundance plasma proteins in 1280 young adults from a multi-ethnic population living in Canada. To our knowledge, this study is the first to explore the relationships between diet, variation in 9p21, and the plasma proteome.

In the present study, individuals homozygous for the 9p21 risk allele (GG genotype) who exhibited low adherence to a Prudent dietary pattern had significantly higher concentrations of Complement C4-γ compared to those with one or no copies of the risk allele. This gene–diet interaction suggests that higher adherence to a Prudent diet may attenuate the proteomic expression of this cardiometabolic risk marker, even among those genetically predisposed. Complement C4-γ has been implicated in the pathogenesis of atherosclerosis, ischemic stroke, obesity, and other clinical phenotypes that contribute to elevated CVD risk [49]. Given the central role of CVD in limiting both lifespan and healthspan [4,50,51], our findings highlight a potential mechanistic pathway through which diet quality may attenuate genetic risk and support functional longevity. Specifically, the 9p21 locus—one of the most well-established genetic markers for CVD—has been shown to be underrepresented in centenarian populations across multiple studies. This suggests that carriers of the high-risk genotype may be less likely to reach extreme old age unless modifying factors, such as protective dietary patterns, are in place. For instance, depletion of 9p21 risk alleles has been observed in Ashkenazi Jewish centenarians [4], and variants at this locus have been linked to cellular senescence and exceptional longevity in diverse cohorts [4,52,53,54]. These findings reinforce the notion that the early adoption of micronutrient-dense, anti-inflammatory dietary patterns may play a critical role in offsetting genetic predispositions and enhancing long-term cardiometabolic resilience [55,56].

Given the long latency period—often spanning several decades—between early exposures and the clinical manifestation of CVD, identifying predictive biomarkers in young adulthood is critical for effective prevention. Copenhaver et al. demonstrated that elevated Complement C4-γ concentrations were associated with impaired endothelial function in healthy young adults, independent of obesity. This suggests a potential role for C4-γ in initiating vascular dysfunction well before overt disease develops. However, their study did not consider the contribution of genetic susceptibility, such as 9p21 variants, which may interact with early-life exposures such as diet to influence long-term risk [57].

Despite emerging evidence, few studies have specifically examined the role of Complement C4-γ in predicting CVD risk among young adults. Xing et al. reported that elevated levels of Complement C4-γ were significantly associated with future incidences of CVD and cerebrovascular events, though their findings were based on a cohort of patients undergoing hemodialysis [58]. Similarly, Nilsson et al. observed a strong association between increased Complement C4-γ concentrations and CVD risk factors, but their study population consisted of 70-year-old individuals [59]. These findings highlight the importance of investigating Complement C4-γ in younger, otherwise healthy populations to assess its relevance as an early biomarker in immune and inflammatory pathways that may precede overt disease and contribute to long-term health trajectories.

Several studies have been conducted investigating the association between Complement C4-γ protein levels and age-related diseases, including Alzheimer’s and CVD [26,60]. Complement C4 is integral to the classical and lectin complement pathways, where it enhances pathogen clearance and promotes inflammation through its cleavage products, Complement C4-γ and Complement C4-β [38,61]. Complement C4-γ plays a critical role in the immune system’s response to infection and inflammation. Chronic low-grade inflammation often leads to the sustained elevation of these inflammatory markers, along with other positive acute-phase proteins such as C-reactive protein (CRP), which are commonly used to assess clinical inflammation. The combined effect of elevated Complement C4-γ and CRP reflects ongoing immune activation, contributing to the persistence and progression of chronic inflammatory diseases [62].

We identified a robust interaction between the 9p21 risk genotype, adherence to a Prudent dietary pattern, and circulating levels of Complement C4-γ—suggesting that dietary quality may play a critical role in modulating genetically mediated inflammatory responses. The Prudent pattern, distinguished by a high intake of fruits, vegetables, and micronutrient-dense whole foods, served as an established indicator of a health-promoting dietary behavior within this cohort. Diets such as these, including the Mediterranean diet and the dietary approaches to stop hypertension (DASH) diet, have been shown to be preventative against CVD [57,58], which may be due to the many different protective factors, such as the antioxidant and anti-inflammatory characteristics of the vitamins, minerals, and other bioactives these foods contain [63]. In our post hoc analyses of the micronutrient composition of the Prudent dietary pattern, we observed strong correlations between dietary adherence and several key micronutrients, including vitamins A, C, D, and E, as well as potassium, folate, and magnesium (Table 4). These micronutrients—predominantly derived from plant-based, minimally processed foods—are recognized for their antioxidant and anti-inflammatory properties. The strength of these associations supports the biological plausibility of our findings, suggesting that adherence to a micronutrient-rich dietary pattern may influence complement activation pathways, as reflected in circulating C4-γ levels, particularly among individuals homozygous for the 9p21 GG risk allele. Furthermore, a higher Western dietary pattern in the current study consisted of those with high saturated fat, high sodium, and high sugar intake [31]. We did not find any association between the Western dietary pattern and plasma proteins associated with elevated CVD risk. This finding is consistent with another study that found that while Prudent dietary patterns attenuated the 9p21 genetic risk of CVD, Western dietary patterns had little effect [64]. However, other studies have shown that high fat and processed diets contribute to the onset of CVD, as well as other diseases such as diabetes [65].

Our findings reveal the novel and distinct associations between the 9p21 risk genotype, plasma protein expression, and dietary quality in a multi-ethnic cohort of healthy young adults. While previous studies have reported associations between 9p21 variants, diet, and cardiovascular events, those investigations have largely focused on older populations with established disease, where underlying comorbidities and long-standing metabolic alterations may confound the interpretation of gene–environment interactions. By contrast, the present study captures these relationships in early adulthood, offering a unique window into the molecular signatures that may precede overt clinical manifestations and providing a clearer insight into the modifiable pathways relevant to prevention [64].

To our knowledge, this study is the first to integrate micronutrient-level resolution within a validated dietary pattern framework while evaluating gene–diet interactions on plasma proteomic profiles. While the field of nutrigenomics has advanced significantly in exploring nutrient–gene relationships—particularly in lipid metabolism and cardiometabolic disease—most prior work has focused on single-nutrient models or broad dietary indices without resolving the micronutrient signatures that may drive these effects [62,66]. Our approach addresses this gap by identifying the specific micronutrients that align strongly with adherence to a Prudent dietary pattern and exploring how these interact with the 9p21 risk locus to shape proteomic profiles in young adults. Notably, magnesium, potassium, Folate, and vitamins A, C and E demonstrated highly significant associations with Prudent dietary adherence (*p* < 1 × 10⁻^10^), consistent with their known roles in oxidative stress, inflammation, and vascular health [67]. Notably, sodium—often emphasized as a harmful nutrient—did not exhibit a strong inverse association with Prudent adherence. This may reflect both limitations in capturing dietary sodium via self-report methods [68], and more importantly, a broader implication that in this cohort of relatively healthy young adults, it was the absence of protective, nutrient-dense foods, rather than the excess harmful ones, that most clearly distinguished risk, as seen by a lack of an association with the Western dietary pattern in this and our prior publications related to 9p21 risk factors.

Another nuance highlights a unique contribution of our study: by mapping the nutrient architecture of dietary patterns with gene–plasma protein interactions, we offer a more mechanistic and translatable framework to understand how composite nutrient exposures—not just excesses or deficiencies—can influence early markers of chronic disease risk. This aligns with current precision nutrition paradigms that prioritize synergy, food matrix effects, and biological relevance over isolated nutrient analysis [69,70].

This study marks another novel contribution to precision nutrition and aging science by integrating genomic, dietary, and proteomic data in a large, multi-ethnic cohort of healthy young adults—a population rarely studied at this mechanistic depth. It is, to our knowledge, the first investigation to demonstrate that adherence to a micronutrient-rich dietary pattern can significantly modify proteomic expression linked to homozygosity at the 9p21 cardiovascular risk locus, a variant strongly implicated in premature CVD, cellular senescence, and reduced lifespan. This nutrient-level granularity enabled us to elucidate the plausible biological mechanisms for the observed gene–diet–protein interactions, lending specificity to our interpretation and highlighting the power of micronutrient profiling within dietary pattern frameworks.

In identifying Complement C4-γ as the protein most strongly modified by a gene–diet interaction, we propose a specific mechanistic pathway by which dietary quality may modulate genetic susceptibility to early inflammatory and vascular dysregulation. This finding is further supported by prior studies linking this protein to poor endothelial function, atherosclerosis, and stroke. By using proteomics as an intermediate molecular phenotype, this study bridges the gap between genomic risk, dietary exposure, and clinical endpoints, advancing the utility of high-throughput omics platforms in early life-stage prevention science. Moreover, the relevance of this work extends beyond proteomics. The 9p21 locus has been previously linked to the epigenetic regulation of CDKN2A/B and non-coding RNA ANRIL, which are associated with cellular senescence and tissue remodeling. Future work integrating transcriptomic and epigenomic data could build on this model to reveal coordinated molecular responses to dietary exposures—a critical direction for advancing life-course precision nutrition strategies aimed at healthy aging.

As with any cross-sectional study, our findings are subject to limitations, including the inability to infer causality and our reliance on self-reported dietary intake. Although the use of a validated food frequency questionnaire and rigorous statistical adjustments strengthened internal validity, residual confounding was always possible. Our proteomics panel, while rigorously validated, was targeted and not comprehensive; future studies with untargeted or epigenetic profiling may yield additional insights. Nevertheless, the study’s novel design, robust sample size, multiomics integration, and focus on a young, ethnically diverse population collectively supported the relevance and impact of its findings.

## 5. Conclusions

Altogether, this work presents a rare and scalable model for early-stage precision nutrition research, with the potential to inform more targeted, cost-effective clinical trials. Its integration of genomic, dietary, and proteomic insights—alongside system-level nutrient deconstruction—offers a significant advancement in our understanding of how diet can modify inherited risk and promote functional longevity well before the onset of chronic disease.

## Figures and Tables

**Figure 1 nutrients-17-01398-f001:**
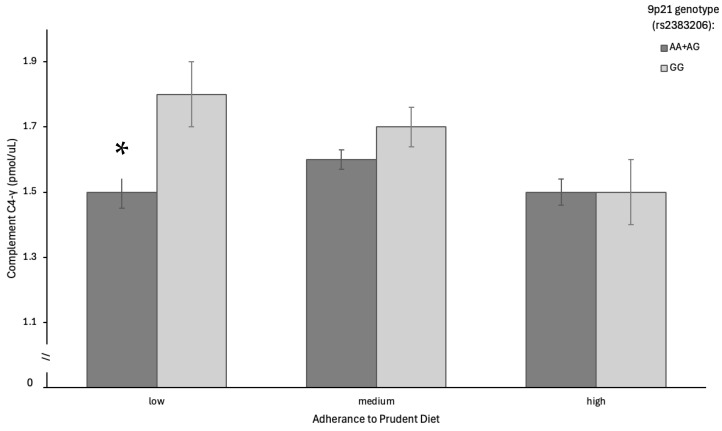
Gene–diet interaction effects on Complement C4-γ. A low Prudent dietary score was associated with a significantly higher concentration of Complement C4-γ in the high genetic risk group (GG) compared to the low genetic risk group (AA+AG) (*p* = 0.02). * Indicates significantly different between AA+AG and GG genotypes in the low diet adherence group.

**Table 1 nutrients-17-01398-t001:** Subject characteristics according to 9p21 genotype (rs2383206, n = 1280).

Characteristic	AA and AG (n = 958)	GG (n = 322)	*p*-Value
Age (Years)	22.7 ± (2.5)	22.5 ± (2.5)	0.32
Body Mass Index (kg/m^2^) *	23.0 ± (3.7)	23.1 ± (3.7)	0.56
Waist Circumference (cm)	74.8 ± (9.5)	75.7 ± (10.3)	0.12
Systolic Blood Pressure (mm Hg)	114.3 ± (12.2)	115.5 ± (12.0)	0.13
Diastolic Blood Pressure (mm Hg)	69.3 ± (7.9)	69.6 ± (9.0)	0.56
Glucose (mmol/L)	4.8 ± (0.4)	4.8 ± (0.4)	0.20
Insulin (pmol/L) *	45.5 ± (38.3)	49.9 ± (38.4)	0.03
CRP (mmol/L)	0.9 ± (2.3)	1.0 ± (2.0)	0.50
TG (mmol/L) *	0.9 ± (0.6)	0.9 ± (0.6)	0.61
HDL (mmol/L)	1.5 ± (0.4)	1.5 ± (0.4)	0.11
LDL (mmol/L)	2.3 ± (0.6)	2.2 ± (0.6)	0.05
Prudent Diet Score	2.9 ± (1.6)	2.8 ± (1.6)	0.21
Eastern Diet Score	1.2 ± (1.0)	1.1 ± (0.8)	0.03
Western Diet Score	1.3 ± (1.0)	1.3 ± (1.0)	0.39
Females, N (%)	593 (76.9)	178 (23.1)	
Males, N (%)	365 (71.7)	144 (28.3)	
Caucasian, N (%)	391 (72.1)	151 (27.9)	
East Asian, N (%)	396 (79.7)	101 (20.3)	
South Asian, N (%)	102 (66.7)	51 (33.3)	
Other, N (%)	69 (78.4)	19 (21.6)	

Values are mean ± (standard deviation). *p*-values are for comparison between the genotypes AA and AG combined, and GG, using multivariate linear regression model. CRP: C-reactive protein, HDL: high-density lipoprotein, LDL: low-density lipoprotein, TG: triglyceride. * Variables were log-transformed to normalize distribution for model building.

**Table 2 nutrients-17-01398-t002:** Average protein concentration by 9p21 genotype (rs2383206) and Prudent dietary score (n = 1280).

	Prudent Dietary Score			*p*-Value		
Protein and Genotype	Low	Medium	High	Diet	Gene	Interaction
α_1_ Antichymotrypsin				0.20	<0.01	0.02
AA and GA	3.2 ± 0.7	3.4 ± 0.8	3.4 ± 0.8			
GG	3.6 ± 0.8	3.5 ± 0.9	3.6 ± 0.7			
Complement C4 β chain				0.45	<0.01	0.05
AA and GA	1.4 ± 0.5	1.4 ± 0.5	1.4 ± 0.5			
GG	1.6 ± 0.6	1.4 ± 0.6	1.4 ± 0.6			
Complement C4 γ chain				0.59	<0.01	0.02
AA and GA	1.5 ± 0.6	1.6 ± 0.6	1.5 ± 0.5			
GG	1.8 ± 0.7	1.7 ± 0.7	1.5 ± 0.7			
Complement C9				0.02	0.09	0.03
AA and GA	2.5 ± 0.9	2.7 ± 0.8	2.8 ± 0.9			
GG	2.8 ± 0.7	2.8 ± 1.0	2.8 ± 0.7			
Fibrinogen α chain				0.08	0.91	0.03
AA and GA	10.7 ± 4.5	12.2 ± 7.0	11.9 ± 5.8			
GG	11.6 ± 3.5	11.9 ± 4.4	11.8 ± 3.5			
Fibrinopeptide A				0.09	0.55	0.02
AA and GA	6.4 ± 2.2	7.1 ± 3.5	7.0 ± 2.7			
GG	7.1 ± 2.2	7.1 ± 2.5	7.0 ± 1.9			
Hemopexin				0.04	0.01	0.01
AA and GA	9.5 ± 1.8	9.7 ± 2.0	10.0 ± 2.3			
GG	10.2 ± 2.0	9.9 ± 2.6	10.6 ± 2.1			
Serum amyloid P-component				0.18	<0.01	<0.01
AA and GA	0.4 ± 0.1	0.4 ± 0.2	0.4 ± 0.1			
GG	0.5 ± 0.1	0.5 ± 0.2	0.4 ± 0.1			

Adjusted for age, ethnicity, log BMI, sex, and physical activity. Values are the mean protein concentration ± standard deviation, listed in order of increasing Prudent dietary scores; the *p*-values are for linear regression models with protein concentrations as the dependent variable and the Prudent dietary pattern and binary genotype as the main determinants of interest, as well as diet–gene as the interaction term.

**Table 3 nutrients-17-01398-t003:** Summary of 9p21 genotype and Prudent dietary pattern interactions on significant plasma proteins.

Protein	Primary Function	Direction of Association	Interaction Interpretation	Selected References
α1 Antichymotrypsin	Serine protease inhibitor; modulates inflammation	* ↑ in GG across diet adherence levels	May reflect persistent low-grade inflammation among GG carriers regardless of dietary pattern	[36,40]
Complement C4 β chain	Classical complement pathway protein	↑ in GG at low diet adherence → at higher adherence	Suggests increased innate immune activity in GG carriers mitigated by Prudent diet	[41,42]
Complement C4 γ chain	Immune-related isoform of C4	↑ in GG at low adherence ↓ at high adherence	Strong diet–genotype interaction; Prudent diet buffers genotype-related inflammatory activation	[5,6,43]
Complement C9	Forms membrane attack complex (MAC)	↑ in GG at low adherence → at medium/high adherence	May reflect subclinical complement activation modifiable by dietary antioxidants	[43,44,45]
Fibrinogen α chain	Precursor to fibrin in clot formation	↑ in GG at low adherence → at higher adherence	Suggests vascular activation in GG carriers; diet may attenuate thrombotic risk	[38,39]
Fibrinopeptide A	Thrombin-cleaved peptide; associated with coagulation	↑ in GG at low adherence → at medium/high adherence	Indicates diet-modifiable pro-thrombotic signaling in high-risk genotype	[9]
Hemopexin	Heme-binding antioxidant protein	↑ in GG across diet; slight ↑ with diet in both genotypes	Reflects enhanced antioxidant demand in GG carriers, with additive benefits from Prudent diet	[10,46]
Serum amyloid P-component	Pentraxin protein; linked to tissue remodeling and amyloid stability	↑ in GG across all diet levels → no modulation by diet	Suggests genotype-driven inflammatory profile unaffected by dietary adherence	[46,47,48]

* Up arrow (↑) indicates higher concentrations of the protein. Down arrow (↓) indicates lower concentrations. Horizonal arrow (→) indicates no difference in protein concentration.

**Table 4 nutrients-17-01398-t004:** Micronutrient correlations with Prudent dietary patterns.

Micronutrient Intake Derived from FFQ	r	*p*-Value
Vitamin A	0.57	2.7 × 10^−45^
Vitamin C	0.31	2.3 × 10^−13^
Vitamin D	0.07	0.12
Vitamin E	0.46	1.6 × 10^−28^
Folate	0.31	5.7 × 10^−13^
Sodium	0.13	0.004
Potassium	0.55	6.8 × 10^−43^
Magnesium	0.62	3.3 × 10^−58^

FFQ; Food Frequency Questionnaire. Correlation coefficients (r) and *p*-values represent associations between dietary patterns and micronutrient intake. Positive correlations indicate higher micronutrient intake with greater adherence, while negative correlations reflect inverse relationship.

## Data Availability

All the data for this study will be made available upon reasonable request to the corresponding author.

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
