# Peer review of "Multiomics Analysis of a Micronutrient-Rich Dietary Pattern and the Aging Genotype 9p21 on the Plasma Proteome of Young Adults"

_nutrients, 2025, doi:10.3390/nu17081398_

Round 1
Reviewer 1 Report
Comments and Suggestions for Authors
The authors studied the association of a 9p21 risk locus, diet type and plasma biomarkers. A new finding is that there is an association between the risk allele and increased levels of complement C4gamma on a low prudent dietary pattern but not on high dietary pattern.
Majot Comments:
The authors performed two-way ANOVA and reported effects of diet, genotype and their interaction. Of the 54 plasma proteins quantified, 8 exhibited a statistically significant interaction. However, the authors then only discuss Complement C4-gamma that shows the expected pattern/significance in posthoc test (Figure 1). The authors should also discuss/interpret the interaction effect (otherwise: why did they test for an interaction?).
Which posthoc test was performed in Figure 1?
The authors should show data for all 54 proteins examined (e.g. in Supplementary Informations).
The authors state and show in Table 1 that the proportion of ethnic groups differed between genotype groups: did the authors examine whether this could have an effect on the plasma proteome data?
There are also some statistically significant differences between genotype groups (insulin, LDL): did the authors examine if these could have an effect on the outcome of the study?
Minor Comments:
Cholesterol is only mentioned in result text line 289; but in order to be consistent, it should also be included in Table 3, though it is not a micronutrient, but neither is sodium (maybe first column of Table 3 should be renamed).
Several Typo in discussion: Table 3: “Microutrient”; Line 353: “composoion”; Line 359: “propoeties”
Line 362: “.... and high [36].” (sentence is incomplete)
Author Response
April 14th, 2025
Dear Dr. Vazquez,
We are pleased to submit our revised manuscript, entitled “Multiomics Analysis of a Micronutrient-Rich Dietary Pattern & Aging Genotype of 9p21 on the Plasma Proteome of Young Adults,” for continued consideration in the Special Issue Recent Advances in Nutrigenomics and Nutrigenetics in Nutrients.
This study offers important new insights into gene-diet interactions by examining how adherence to dietary patterns modifies the impact of the 9p21 pleiotropic risk locus and accelerated biological aging on circulating proteomic biomarkers. Drawing on data from a young, multiethnic cohort, we integrated plasma proteomics with genetic and dietary analyses to address a critical gap in understanding early-life gene-environment interplay related to chronic disease risk. Our findings suggest that adherence to a micronutrient-rich prudent dietary pattern may attenuate the effects of the 9p21 risk genotype on cardiometabolic protein signatures—particularly complement C4-γ—highlighting a potential mechanism by which targeted dietary interventions may mitigate inherited risk. These results have strong translational relevance for advancing personalized nutrition strategies in genetically susceptible populations.
We thank you and the reviewers for your thoughtful input and consideration. We have addressed all suggestions and questions in a point-by-point rebuttal below and accompanying updated manuscript.
We look forward to the opportunity to contribute to this timely and important Special Issue.
Sincerely,
On behalf of all authors
Response to Editor
We thank Dr Vazquez for the opportunity to revise and resubmit our manuscript. We appreciate the constructive feedback provided by the reviewers and have carefully addressed each of their comments in a point-by-point manner.
In accordance with the journal’s instructions:
- References: We reviewed all references to ensure they are directly relevant to the content of the manuscript. Any outdated or non-essential references have been removed or replaced with more current and appropriate citations.
- Manuscript Revisions: All revisions to the manuscript have been clearly highlighted to facilitate the review process. This includes both textual edits and updates to figures and tables, where applicable.
- Point-by-Point Response: A detailed response to each reviewer comment has been provided below. We have outlined the exact changes made to the manuscript in response to each point and indicated where these changes can be found.
- Suggested References: Where reviewers recommended additional references, we critically evaluated each suggestion. Relevant citations that strengthen the manuscript have been incorporated, while others were not included with justification provided in our responses.
- Unaddressed Comments: We were able to address all reviewer comments. No additional appeals or unresolved points remain at this time.
We believe the manuscript has been significantly improved as a result of the reviewers’ input and hope that the revised version will be suitable for publication in your journal. We thank you again for your consideration and look forward to your prompt feedback.
Response to Reviewer 1
We thank Reviewer 1 for their thoughtful and constructive feedback. We appreciate the time taken to review our manuscript and have carefully addressed each comment below.
- The authors performed two-way ANOVA and reported effects of diet, genotype and their interaction. Of the 54 plasma proteins quantified, 8 exhibited a statistically significant interaction. However, the authors then only discuss Complement C4-gamma that shows the expected pattern/significance in posthoc test (Figure 1). The authors should also discuss/interpret the interaction effect (otherwise: why did they test for an interaction?).
Thank you for raising this important point. We agree that a more thorough discussion of the significant interaction effects is with other proteins is warranted. In response, we have revised the Results and Discussion sections to include a detailed analysis of all eight proteins exhibiting significant genotype-diet interactions, as presented in Table 2 and 3.
While Complement C4-γ demonstrated the clearest and most biologically interpretable interaction, showing an attenuated GG genotype effect with higher adherence to a prudent dietary pattern, we now provide further context for the remaining seven proteins. For instance, Hemopexin, Serum amyloid P-component, and Complement C4 β chain displayed modest, albeit less pronounced, differences across genotype and diet strata. Conversely, Fibrinogen α chain and Fibrinopeptide A exhibited minimal variation with dietary adherence, suggesting that the statistically significant interactions may reflect subtle, scale-dependent shifts rather than robust biological effects.
We have clarified that while only Complement C4-γ aligned fully with our initial hypothesis, the identification of seven additional proteins with significant interactions highlights the potential of proteomic profiling to reveal early molecular responses to gene-diet interactions. We now emphasize that these findings should be considered hypothesis-generating, guiding future targeted investigations. We thank the reviewer for their suggestion which provide an opportunity to include this data for a more rich article and discussion points.
- Which posthoc test was performed in Figure 1?
We appreciate the reviewer’s attention to statistical detail. In Figure 1, we employed a one-way ANOVA to assess differences across the tertiles of the prudent dietary pattern. Following a significant ANOVA result, we conducted Tukey's Honest Significant Difference (HSD) test for post hoc pairwise comparisons. This method was chosen for its robustness in controlling the family-wise error rate when performing multiple comparisons. We have now clarified this in the figure legend and the Statistical Analysis section of the Methods to enhance transparency.
- The authors should show data for all 54 proteins examined (e.g., in Supplementary Information).
Thank you for this suggestion. We have added a new Supplementary Table (Supplementary Table 1) that includes descriptive statistics, p-values, and interaction terms for all 54 proteins analyzed in the study. A reference to this table has been added in the Results section.
- The authors state and show in Table 1 that the proportion of ethnic groups differed between genotype groups: did the authors examine whether this could have an effect on the plasma proteome data?
Thank you for this thoughtful observation. We agree that population stratification is an important consideration in gene-environment interaction studies. As shown in Table 2, we included self-identified ethnicity, along with age, sex, BMI (log-transformed), and physical activity, as covariates in all primary models assessing gene–diet interactions. Additionally, we conducted preliminary sensitivity analyses stratified by sex and by ethnocultural background. These analyses revealed no significant modification of the genotype–diet–protein associations by these factors, which justified collapsing the groups for final analyses. The interaction between 9p21 genotype and prudent dietary adherence on Complement C4-γ, and other significant proteins, remained statistically robust across these models.
We have now clarified this more explicitly in the revised Methods and Results sections.
- There are also some statistically significant differences between genotype groups (insulin, LDL): did the authors examine if these could have an effect on the outcome of the study?
Thank you for raising this important point. The observed differences in insulin levels between genotype groups are consistent with prior work by our group, as described in introduction and supported by references 8 and 17. In earlier studies published in 2018 and 2022, we characterized the relationship between the 9p21 genotype and cardiometabolic markers, both independently and in combination with dietary patterns. These findings established a foundational understanding of genotype–phenotype associations within this cohort. The present study builds upon that work by extending our investigation into proteomic markers as part of a multi-omics approach. As such, cardiometabolic differences are expected features of the genotype groups rather than confounding variables, and our aim here was to examine whether these established genetic differences manifest through proteomic pathways and are modifiable by diet. The differences observed in Table 1 for LDL were borderline statistically significant and not clinically meaningful. Furthermore, the Table 1 data are unadjusted descriptive group differences and LDL was not statistically different in the adjusted analyses in the subsequent adjusted models (ie adjusted for age, sex and BMI). We have clarified this in the revised Discussion section.
- Cholesterol is only mentioned in result text line 289; but in order to be consistent, it should also be included in Table 3, though it is not a micronutrient, but neither is sodium (maybe first column of Table 3 should be renamed).
We appreciate this insightful comment and have clarified an important and novel aspect of our study. Cholesterol was not included in Table 3 because it is not classified as an essential nutrient, nor does it fall under standard macro- or micronutrient categories, given its endogenous synthesis and non-essential status in the diet. In contrast, sodium is an essential mineral and is therefore appropriately categorized as a micronutrient. We have updated the title of Table 3 (now renumbered as Table 4 due to the addition of a new protein-focused table) to more clearly reflect this distinction. In this cohort, cholesterol intake was not a significant determinant of prudent dietary adherence and did not contribute meaningfully to the dietary patterns under study. Importantly, our analysis was structured to highlight biologically relevant nutrient patterns that are not typically assessed in isolation in clinical or epidemiological settings. Rather than focusing on singular nutrients, we intentionally emphasized micronutrient combinations that demonstrated the strongest associations with dietary adherence and were mechanistically plausible in the context of gene–diet interactions. This approach reflects a broader move in nutrition science toward understanding synergistic effects of nutrient patterns, particularly those that may modify genetically driven risk. In response to this suggestion, we have now better emphasized in the manuscript that one of the novel aspects of our study lies in its micronutrient-level resolution of dietary pattern associations. Rather than examining individual nutrients in isolation, or focusing on macronutrients done in the past, our goal was to characterize patterns of micronutrient intake that directionally aligned with, or diverged from, prudent dietary adherence, especially in the context of gene–diet interactions. Indeed, several micronutrients—such as magnesium, potassium, folate and vitamin A and vitamin C, vitamin E and others—demonstrated extremely strong associations with dietary pattern adherence, with p-values reaching levels of 10⁻¹⁰ or lower. Importantly, sodium showed a significant association but not nearly as strong of these protective micronutrients, underscoring its value in helping to characterize overall dietary quality. We have made this clearer in the manuscript discussion. We greatly appreciate this reviewer’s suggestions, which prompted us to further emphasize the unique findings of our study.
- Several typos in discussion: Table 3: “Microutrient”; Line 353: “composoion”; Line 359: “propoeties”
Thank you for catching these typographical errors. These have been corrected in the revised manuscript.
- Line 362: “.... and high [36].” (sentence is incomplete)
This sentence has now been revised for clarity and completeness. We regret the oversight and thank the reviewer for noting it.
We hope the revisions and additional analyses address this reviewer’s concerns and enhance the clarity, completeness, and scientific value of our manuscript. Thank you once again for your thoughtful review.
Reviewer 2 Report
Comments and Suggestions for Authors
The objective of this study was to explore the impacts of dietary patterns and the 9p21 genotype on the plasma proteome of young adults for early CVD prevention. 1,280 multi - ethnic young adults were recruited. Dietary intake was assessed, dietary patterns determined, gene SNP sites analyzed, and associated proteins quantified. The 9p21 genotype significantly interacted with the Prudent dietary pattern for eight proteins, with complement C4-γ significantly affected. The Prudent dietary pattern was linked to multiple nutrient intakes. This study, the first of its kind, offers a basis for dietary interventions to reduce CVD risk. The outlook is very promising.
Overall, the article is well organized and its presentation is good. However, some problems still need to be improved:
- It is recommended that the references cite more literature from the last five years.
- In the Introduction, when introducing the plasma proteomics method, briefly mention its application cases and effects in similar studies to enhance the persuasiveness of the method selection.
- In the Materials and Methods section, clarify whether the questionnaire results were calibrated to reduce possible measurement errors.
- In this study, how is the lack of association between Western dietary patterns and plasma proteomes, which are linked to elevated CVD risk, effectively demonstrated? It is recommended to present more persuasive arguments to strengthen this aspect.
Author Response
Response to Reviewer 2
The objective of this study was to explore the impacts of dietary patterns and the 9p21 genotype on the plasma proteome of young adults for early CVD prevention. 1,280 multi - ethnic young adults were recruited. Dietary intake was assessed, dietary patterns determined, gene SNP sites analyzed, and associated proteins quantified. The 9p21 genotype significantly interacted with the Prudent dietary pattern for eight proteins, with complement C4-γ significantly affected. The Prudent dietary pattern was linked to multiple nutrient intakes. This study, the first of its kind, offers a basis for dietary interventions to reduce CVD risk. The outlook is very promising.
Overall, the article is well organized and its presentation is good.
We thank Reviewer 2 for the thoughtful and constructive comments provided. Below, we offer point-by-point responses, each followed by a description of the revisions made to the manuscript.
- Comment: “It is recommended that the references cite more literature from the last five years.”
We appreciate this helpful suggestion. We have reviewed the manuscript carefully and updated multiple references to include recent publications from 2019 to 2025 that reflect current advances in the fields of nutrigenomics, dietary pattern analysis, proteomics, and 9p21 genotype. Specifically, we have incorporated the following studies: Zhu (2023), Mahdavi (2022), Zheng (2022), Ghoch (2024), Mente (2023), Nielsen (2023), Sapey (2020), Crossley (2019), McMurray (2024), Donkin (2023), Wang (2024), Copenhaver (2020), Xing (2022), Heurich (2024), Wang (2021), Badmion (2019), Diab (2023), Barabási (2019). These references enhance the contemporary relevance and scientific foundation of the manuscript.
- Comment: “In the Introduction, when introducing the plasma proteomics method, briefly mention its application cases and effects in similar studies to enhance the persuasiveness of the method selection.”
Thank you for this important observation. We have revised the Introduction to clarify the rationale for using a targeted proteomics panel consisting of 54 abundant plasma proteins. This panel was selected based on its prior use in large-scale population studies investigating circulating proteins linked to inflammatory, metabolic, and vascular processes. Most importantly, we have previously demonstrated the utility of this panel within the Toronto Nutrigenomics and Health Study (TNHS), where several analyses have confirmed its capacity to detect biologically meaningful variation associated with diet, genetics, and other lifestyle exposures. These findings provide strong internal validation for its use in our current study. At the same time, we acknowledge that this panel covers only a subset of the plasma proteome, and we now explicitly discuss the limitations in terms of external validity and generalizability to broader proteomic platforms.
- Comment: “In the Materials and Methods section, clarify whether the questionnaire results were calibrated to reduce possible measurement errors.”
We thank the reviewer for this important clarification. Dietary intake was assessed using the Toronto-modified Harvard Food Frequency Questionnaire (FFQ), which was specifically developed and validated for use in this multi-ethnic cohort of young adult. The validation study, published by Neilsen et al. in 2023 (Eur J Clin Nutr), demonstrated that the FFQ produced reliable estimates of energy and nutrient intake compared to 3-day food records. The study confirmed acceptable levels of reproducibility and relative validity for key nutrients relevant to cardiometabolic health, including micronutrients and macronutrients. This validation work supports the robustness of our dietary intake data, and the Methods section has been updated accordingly to reflect this.
- Comment: “In this study, how is the lack of association between Western dietary patterns and plasma proteomes, which are linked to elevated CVD risk, effectively demonstrated? It is recommended to present more persuasive arguments to strengthen this aspect.”
We agree that this null finding requires additional contextualization. In the revised Discussion, we now elaborate on the potential explanations for the absence of strong proteomic associations with the Western dietary pattern. Although this pattern was associated with higher intake of saturated fats and added sugars—nutrients commonly linked to cardiovascular risk—our results may reflect the relatively young, asymptomatic status of participants in this cohort. Proteomic alterations linked to deleterious exposures may require more prolonged or cumulative exposure to become detectable. In addition, the specific proteomic panel used in this study may be more sensitive to early protective effects rather than adverse dietary patterns. We now discuss these biological and methodological factors and explicitly note that broader or longitudinal proteomic profiling may be required to capture such associations in future work.
We are grateful to the reviewer for these insightful comments, which have led to important clarifications and improvements in the manuscript.
Round 2
Reviewer 1 Report
Comments and Suggestions for Authors
All critical points raised by this reviewer have been addressed satisfactorily by the authors.